# Interaction Study between ESIPT Fluorescent Lipophile-Based Benzazoles and BSA

**DOI:** 10.3390/molecules26216728

**Published:** 2021-11-06

**Authors:** Thais Kroetz, Pablo Andrei Nogara, Fabiano da Silveira Santos, Lilian Camargo da Luz, Viktor Saraiva Câmara, João Batista Teixeira da Rocha, Alexandre Gonçalves Dal-Bó, Fabiano Severo Rodembusch

**Affiliations:** 1Grupo de Pesquisa em Fotoquímica Orgânica Aplicada, Instituto de Química, Universidade Federal do Rio Grande do Sul, Av. Bento Gonçalves 9500, Bairro Agronomia, Porto Alegre 91501-970, CEP, Brazil; kroetzthais@gmail.com (T.K.); fabiano@ufrgs.br (F.d.S.S.); lilian.luz@ufrgs.br (L.C.d.L.); viktor.saraiva@gmail.com (V.S.C.); 2Departamento de Bioquímica e Biologia Molecular, Centro de Ciências Naturais e Exatas, Universidade Federal de Santa Maria-UFSM, Santa Maria 97105-900, RS, Brazil; pbnogara@gmail.com (P.A.N.); jbtrocha@yahoo.com.br (J.B.T.d.R.); 3Programa de Pós-Graduação em Ciência e Engenharia de Materiais, Universidade do Extremo Sul Catarinense (UNESC), Av. Universitária 1105, Criciúma 88806-000, CEP, Brazil; adalbo@unesc.net

**Keywords:** benzazoles, proton transfer, bovine serum albumin, fluorescence quenching, molecular docking

## Abstract

In this study, the interactions of ESIPT fluorescent lipophile-based benzazoles with bovine serum albumin (BSA) were studied and their binding affinity was evaluated. In phosphate-buffered saline (PBS) solution these compounds produce absorption maxima in the UV region and a main fluorescence emission with a large Stokes shift in the blue–green regions due to a proton transfer process in the excited state. The interactions of the benzazoles with BSA were studied using UV-Vis absorption and steady-state fluorescence spectroscopy. The observed spectral quenching of BSA indicates that these compounds could bind to BSA through a strong binding affinity afforded by a static quenching mechanism (K_q_~10^12^ L·mol^−1^·s^−1^). The docking simulations indicate that compounds **13** and **16** bind closely to Trp134 in domain I, adopting similar binding poses and interactions. On the other hand, compounds **12**, **14**, **15**, and **17** were bound between domains I and III and did not directly interact with Trp134.

## 1. Introduction

Albumin is a globular, water-soluble serum protein and is the most abundant protein in the blood. It binds many ligands including fatty acids, trace elements, steroids, thyroid hormones, hemin, calcium, and other molecules [1,2]. This protein is known to play an essential role in the transport and disposition of endogenous and exogenous compounds in blood [3]. Binding studies of therapeutic agents with plasma proteins, particularly serum albumin, provide valuable information on absorption, transportation, distribution, metabolism, and efficacy [4]. Thus, bovine serum albumin (BSA) is frequently chosen as a model for drug–protein interaction studies because of its low cost, availability, and homology with human serum albumin (HSA) [5,6]. BSA exhibits intrinsic fluorescence, mostly due to the indole group of tryptophan (Trp), and the interaction with small molecules can cause fluorescence quenching [7]. In addition, fluorescence titration has been widely applied to study the binding affinity of small molecules to these biomacromolecules, allowing their non-intrusive evaluation under physiological conditions [8]. In this context, 2-(2′-hydroxyphenyl)benzazole-based compounds are privileged compounds owing to their structural and electronic properties [9,10,11]. These fluorophores are able to transfer protons in the excited state due to an intramolecular hydrogen bond between the hydroxyl group (proton donor) and the azolic nitrogen (acceptor group) [12,13]. The photoacidity/photobasicity, which is influenced by bond distance and angles, induces a fast proton transfer to produce a keto phototautomer, which decays and emits fluorescence with a large Stokes shift (approx. 10000 cm^−1^) [14,15]. These photophysical features allow these compounds to be applied as pH sensors [16,17,18,19], WOLEDs [20,21,22,23,24,25], optical sensors [26,27,28], bioimaging [29,30,31,32,33,34,35], and lipid probes [36,37,38,39,40].

In this study, the binding affinity for BSA of photoactive lipophilic amines based on the benzazolic fluorophore, comprising benzoxazole and benzothiazole compounds, was evaluated. First, their electronic properties were investigated in phosphate buffer solution (PBS) in both the ground and excited states by UV-Vis absorption and steady-state fluorescence emission, respectively. Their application as optical sensors to detect proteins in PBS was successfully explored using BSA as a model. These compounds were chosen due to their high photostability and absent inner-filter effect, which could be interesting for association studies. In addition, the presence of different alkyl chains could also affect their interaction with BSA by suppression studies, provided by hydrophobic interactions with the macromolecule. Finally, molecular modeling studies were performed to better visualize the binding mode with BSA.

## 2. Results and Discussion

### 2.1. Synthesis and Photophysical Characterization

The fluorescent secondary amines used in this study are similar to those reported previously [41]. Generally, the reaction of the formyl derivatives **1**–**2** [14] with an equimolar quantity of amines **3**–**5** in dry isopropanol/acetic acid (catalyst) under reflux yielded the respective imines **6**–**11** (Figure 1). In this step, the solvent was evaporated to dryness to afford **6**–**11** in a ~90% crude yield, and no further purification was required to proceed with the next step. The desired amines were obtained by reducing the imines with sodium borohydride in a 1:2 (imine:NaBH_4_) ratio. The final products were obtained after purification via column chromatography in satisfactory yields (~70%) using ethyl acetate as the eluent.

The photophysical investigation was carried out in PBS (pH 7.2) at a concentration of 10^−5^ M. The characterization data from the electronic ground and excited state are summarized in Table 1. As shown in Figure 1, the benzazoles exhibit absorption maxima in the UV region, as already observed in organic media [41]. The benzothiazoles produced red-shifted absorption maxima compared to their benzoxazole analogs, which can be explained by the better electron delocalization transferred from the sulfur to the oxygen [12]. Surprisingly, both derivatives with short alkyl chains (**12** and **15**) exhibited absorption maxima at high energies (~330 nm) despite the compounds with longer lipophilic chains, indicating that the alkyl chain somehow plays a role in their electronic properties in the ground state. In addition, for compounds **13**–**14** and **16**–**17**, broad absorption was observed in the visible region of the spectra. This feature could be a result of scattering and may be an indication of aggregation of these compounds in PBS (pH 7.2), due to the presence of the long alkyl chains in these compounds, which are extremely hydrophobic. Significantly, the similarity in shape and maxima wavelength compared to those of organic solvents, as well as the absence of additional red-shifted bands in PBS, indicates the absence of ionized species in the ground state for derivatives in this media [42,43].

Finally, the UV-Vis absorption spectra were used to calculate the experimental extinction coefficients ε from the oscillator strengths *f*_e_ using the Strickler–Berg relationship represented by Equation (1) [44]. In this equation, the integral is related to the area of the absorption maxima from a plot of *ε* (M^−1^·cm^−1^) vs. $\bar{v}$ (wavenumber, cm^−1^), correlated to a single electron oscillator.
(1)$$f_{e}\approx 4.3\times 10^{-9}\int^{\;}\varepsilon d\bar{v}$$

The radiative rate constant, *k*_e_^0^, can also be obtained using ε by applying Equation (2), where $\bar{v}$_0_ is the absorption maxima (cm^−1^) [45].
(2)$$k_{e}^{0}\approx 2.88\times 10^{-9}{\bar{v}}_{0}^{2}\int^{\;}\varepsilon d\bar{v}$$

The calculated molar absorptivity values (*ε*, ~10^4^ M^−1^⋅cm^−1^), as well as the radiative rate constants (*k*_e_^0^, ~10^8^ s^−1^) indicate spin and symmetry-allowed electronic transitions, which could be related to ^1^π-π* transitions. Similar values were obtained for the same compounds in organic media, as discussed in the literature [41]. Moreover, an almost constant radiative lifetime *τ*_0_ indicates that these benzazoles seem to populate the same excited state after radiation absorption.

The steady-state fluorescence emission spectra of the studied benzazoles are presented in Figure 2. The curves were obtained by exciting the compounds at the absorption maxima (Table 1). In general, all compounds produced the main emission band located above 450 nm with a large Stokes shift. However, the nature of the emission bands seems to be quite different depending on the hydrophobic portion of the molecule, once again indicating that the alkyl chains influence their photophysics. For the benzoxazole derivatives, compound **12** demonstrated a blue-shifted emission (464 nm, Δλ_ST_ 8751 cm^−1^) when compared to analogs **13** (502 nm, Δλ_ST_ 8651 cm^−1^) and **14** (492 nm, Δλ_ST_ 8246 cm^−1^). Considering the Stokes shift values and the emission maxima location [9], we believe that the emission from compound **12** was produced due to its ionized species. Moreover, from the same parameters, the emission from the additional derivatives **13** and **14** could be related to the ESIPT process (tautomeric emission). Benzothiazole derivatives **15–17** produced similar results. Compound **15**, which has a short alkyl chain, emitted fluorescence at 467 nm with a Stokes shift of 8707 cm^−1^. For **16–17**, both the emission maxima (542 nm) and Stokes shifts (~8800 cm^−1^) were observed at higher wavelengths. Thus, the emission produced by compound **15** is probably related to its ionized species [42]. Compounds **16–17** exhibit proton transfer in the excited state, and the emission is a result of this process. In addition, compounds **16** and **17** produced a weak blue-shifted emission (462 nm), which is also believed to be related to their ionized species. The enol or normal emission was disregarded because of the high Stokes shift values related to these emission bands.

### 2.2. BSA Binding Study

Interaction studies were carried out to investigate the affinity of benzazoles **12**–**17** with BSA through fluorescence quenching assays at room temperature (298 K). BSA is composed of a single polypeptide chain of 583 amino acid residues [46] and three domains named I, II, and III, each of which is divided into two subdomains (A and B) [47]. Studies indicate that the main association sites are located in the hydrophobic cavities of the IIA and IIIA subdomains [48]. The BSA structure has two Trp residues, Trp-134 in the first domain, located on the surface of the molecule, and Trp-212 in the IIA subdomain, located in the hydrophobic region of the protein [49,50]. BSA produced an absorption band in the UV-Vis region, with a maximum of around 280 nm, and an intrinsic fluorescence emission with a maximum located between 340–350 nm, depending on the excitation wavelength, owing to the Trp residues [6]. Considering these photophysical properties, the interactions of small molecules with BSA generally lead to the suppression of fluorescence emission, which is a powerful indicator of their interaction mechanisms [3,4,8,51], which are generally classified as dynamic, static, or combined quenching. Static suppression mechanisms are induced by complex formation between BSA and the dye in the ground state. Dynamic suppression is caused by collisions between BSA and the dye in the excited state. In both cases, energy is transferred from the protein to the compound, resulting in the suppression of fluorescence [52]. Thus, to investigate the fluorescence quenching of BSA in the presence of the benzazoles, spectrophotometric titration was performed using different quantities of dye (2–20 μM) and constant BSA concentration (11 μM, 1 mL) in PBS (pH 7.2). This study was performed at room temperature (25 °C), and the suppression percentage was calculated from the emission intensities obtained under excitation at 277 nm. 

Figure 3 shows that the UV-Vis absorption bands of the dyes increase in intensity upon addition of the benzoxazoles **12**–**14** to the solution containing BSA. In addition, the absorption band around 275 nm, related to BSA, also increased due to interference from another interaction that causes fluorescence at a similar wavelength. A similar response was observed for the benzothiazole analogs **15**–**17** (Appendix A). 

BSA emitted intense fluorescence with a maximum at 334 nm under excitation at 277 nm. The fluorescence emission spectra of BSA in the presence of benzazoles **12**–**13** shows an initial decrease of approximately 10% upon the first addition of 2 μM quencher (**12**: 9%, **13**: 7%, and **14**: 8%) and a significant decrease in the fluorescence intensity at 20 μM dye concentration (**12**: 42%, **13**: 50%, and **14**: 41%) with no significant red-shift, suggesting that the interaction site may be close to the Trp residue (Trp-134 or Trp-212). Moreover, these results suggest that there is no perturbation of the microenvironment around the Trp binding site [52]. Similar behavior was observed for the benzothiazole analogs at 2 μM (**15**: 8%, **16**: 6%, and **17**: 5%) and at 20 μM (**15**: 48%, **16**: 53%, and **17**: 41%; Appendix A). Higher values for BSA quenching were obtained for compounds **13** and **15**, which have 12 carbon aliphatic chains, probably due to an optimized balance between hydrophobicity and spatial hindrance.

To identify the main fluorescence quenching mechanism induced by the benzazoles, the Stern–Volmer relationship, described in Equation (3), was applied [53]
(3)$$\frac{F_{0}}{F}=1+K_{\mathrm{sv}}\left[Q\right]=1+k_{q}\tau_{0}\left[Q\right]\;$$
where *F*_0_ is the fluorescence intensity of pure BSA and *F* is the fluorescence intensity in the presence of benzazoles (quencher [Q]). *K*_SV_ is the Stern–Volmer constant, and *k*_q_ is the bimolecular quenching rate constant, which is related to the suppression efficiency. τ_0_ is the fluorescence lifetime of BSA in the absence of the suppressor (6.06 ns) [54]. According to this equation, *K*_SV_ can be obtained from the slope of the linear fit, and *k*_q_ is equal to *K*_SV_/τ_0_. The obtained curves are presented in Figure 4, and the relevant data are summarized in Table 2. The equations for the respective linear fits are listed in Appendix A. High *K*_SV_ values were obtained (10^4^ M^−1^), especially for compounds **13** and **16**, indicating that the benzazoles exhibit moderate to strong interactions with the BSA binding sites [55]. Additionally, the *k*_q_ values (~10^12^ M^−1^·s^−1^) exceed the maximum value for the diffusional collision quenching constant according to the Smoluchowski–Stokes–Einstein theory (*k*_diff_ ≈ 7.40 × 10^9^ M^−1^ s^−1^) [56], which indicates that fluorescence quenching occurred by a static mechanism, in which the formation of a benzazole–BSA conjugate takes place in the ground state.

Thus, given that the reaction was known to proceed via a static mechanism, the binding constant (*K*_A_) and the number of binding sites (*n*) between BSA and the benzazoles were also obtained by applying Equation (4) [57]:(4)$$\log\left(\frac{F_{0}-F}{F}\right)=\log\;K_{A}+n\log\;\left[Q\right]\;$$
where *F*_0_ and *F* represent the fluorescence intensities in the absence and presence of the benzazoles, respectively, and [Q] is the concentration of the quencher (benzazole). The respective curves are shown in Figure 5. Moreover, the standard Gibbs free energy (ΔG^0^) of the benzazole:BSA conjugates was calculated from the *K*_A_ values using Equation (5), where R is the gas constant (1.987 cal· K^−1^·mol^−1^) and T is the temperature (298 K).
ΔG^0^ = −RT ln *K*_A_(5)

The results from the double logarithmic plot of the relationship between the fluorescence intensities of BSA and the benzazoles are summarized in Table 2, where the *K*_A_ values are in the order of 10^4^ M^−1^, suggesting a strong interaction with BSA. The presence of different alkyl chains in the benzazoles is thought to cause variation in the binding constant values, with compounds **13** and **16** exhibiting higher values. In addition, the number of binding sites (*n*) for all compounds was in the range of 0.88–1.31, indicating the presence of one binding site in the interaction with BSA. The calculated Gibbs free energy did not show any trend, with values in the same order of magnitude indicating spontaneous interaction with BSA.

A BSA association study was also performed with different amounts of protein (0–12 μM in PBS) adding these solutions to the respective benzazole dye solution (2 μM in PBS). This study can be observed in Figure 6, using benzazoles **12** and **15** as models. Figure 6a–c depicts their UV-Vis absorption spectra in the absence and presence of BSA. Two distinct regions can be observed, related to the protein (250–300 nm) and the dyes (300–380 nm), depending on the benzazole. Increasing BSA raised the absorption intensity between 250–300 nm, and the dye region remained almost constant. A similar response was observed for all studied compounds (Appendix A). However—and we would like to highlight these findings—based on the emission spectra (Figure 6b–d), the association was observed only for compound **12**, where the fluorescence intensity increased with the BSA amount in solution. All sets of emission spectra and respective plots of fluorescence intensity maxima as a function of BSA concentration are presented in Appendix A.

The first addition of BSA (2 μM) increased the fluorescence intensity by around two times. After the last addition of protein (12 μM) the fluorescence increased by 3.6, and red-shifted the emission maxima 25 nm (Figure 6b). The observed bathochromic shift suggests that the BSA microenvironments are less polar than the PBS due to the hydrophobic groups present in the surface and interiors of the BSA [58]. On the other hand, the parent compound **15** presents an almost constant intensity in the presence of different amounts of BSA (Figure 6d). The additional compounds showed quite different results, sometimes increasing the intensity until reaching a maximum and then decreasing the intensity (compounds **13** and **16**) or randomly varying the intensity (compound **14**), and sometimes even showing fluorescence suppression (compound **17**) (Appendix A). These results indicate that based on the compound’s structure, the interaction with BSA from the point of view of the excited state of the dyes can be a tricky subject. The association (*K*) of compound **12** with BSA was also obtained by the Benesi–Hildebrand equation for 1:1 complex [59]. Figure 7 presents the plot of 1/(*I*−*I*_0_) vs. 1/[BSA]. The linearity of the plot indicates the formation of a 1:1 complex between compound **12** and BSA. The respective binding constant *K* calculated from the slope of the straight line was found to be 1.75 × 10^5^ M^−1^ corroborating with the strong interaction of these compounds with BSA observed by suppression studies.

### 2.3. Molecular Docking

To better understand the interaction between the studied compounds and BSA, and to identify the amino acid residues involved in their interactions, molecular docking simulations were performed. Initially, blind docking was performed for all protein structures. The results indicate that a possible binding site was in a region close to Trp134. BSA contains two Trp residues (Trp134 and Trp213), both of which are responsible for its fluorescence. Trp134 is located in domain I, while Trp213 is in domain II (Figure 8) [60,61].

Second, we performed semi-flexible docking in the region close to Trp134 (with the lateral chain of residues Arg185, Lys136, Lys131, Trp134, Tyr137, and Tyr160 flexible) to improve the interactions between BSA and the compounds. The docking simulations indicate that compounds **13** and **16** bind closely to Trp134 in domain I (Figure 8) and adopt similar binding poses and interactions (Figure 9b,e). These compounds exhibited π–π stacking between the benzene ring and the indole ring of the Trp134 residue (3.8–4.0 Å), hydrophobic interactions between the methyl group and the imidazole ring from the His18 residue and between the carbon chain of the compounds and the carbon chain from the Lys132 residue. They have hydrogen bonds (H-bonds) between the amine moiety and the carboxylic group of the Glu17 residue and between the hydroxyl moiety and the carbonyl group of the Asn158 residue. There is a π–H bond between the benzoxazole (or benzothiazole) group and the amide moiety of the Asn161 residue and a cation–π interaction between the benzene ring of compound **16** and the ammonium group of the Lys131 residue. These direct interactions between the compounds and the indole ring could influence the Trp fluorescence quenching.

However, compounds **12**, **14**, **15**, and **17** were bound between domains I and III (Figure 8) and did not directly interact with Trp134 (Figure 9). In general, these molecules interacted with the Arg458 residue via cation–π interactions and hydrophobic interactions between the benzoxazole (or benzothiazole) group and the lateral chain of the Ala193 residue and between the methyl moiety and the His145 and Pro146 residues. In addition, we observed that the alkyl chains of compounds **14** and **17** were located in different positions, while the carbon chain of **14** interacted with the Phe133 and Lys136 residues, and the carbon chain of **17** exhibited intramolecular hydrophobic interactions (Figure 9c,f). Previous studies have described this region as a BSA-binding site [63].

These observations are consistent with other studies where π–π interactions and H-bonds are involved in BSA–ligand complex stabilization [64]. In addition, the predicted thermodynamic data (∆G_bind_) obtained by molecular docking, differently than observed experimentally, demonstrated that the compounds with short carbon chains (**12** and **15**) exhibited the lowest ∆G_bind_, indicating a more spontaneous interaction with BSA compared with those of other molecules (Table 3).

## 3. Experimental

### 3.1. Photophysical Characterization

Spectroscopic grade solvents were used for fluorescence and UV-Vis absorption spectroscopies. UV-Vis absorption spectroscopy of 10^−5^ M solutions was performed on a Shimadzu UV-2450 spectrophotometer (Shimadzu Corporation, Tokyo, Japan). Steady-state fluorescence spectroscopy was performed using a Shimadzu spectrofluorometer (RF-5301PC). The fluorescence quantum yields (Φ_FL_) were measured at the optical dilute regime with solutions with an absorbance intensity lower than 0.05 and using quinine sulfate (Riedel-de-Haën, Seelze, Germany) in H_2_SO_4_ 1N as the quantum yield standard (Φ_F_ = 0.55) [65].

### 3.2. BSA Interaction Studies

Bovine serum albumin (BSA), lyophilized powder ≥96% (agarose gel electrophoresis) (Sigma-Aldrich, Saint Louis, MO, USA) was used in the protein detection experiments. Phosphate buffered saline (PBS) was prepared from 9.0 g of sodium chloride (NaCl), 1.18 g of monobasic potassium phosphate (KH_2_PO_4_), 4.32 g of anhydrous dibasic sodium phosphate (Na_2_HPO_4_), and 1.0 L of Milli-Q water, resulting in a saline solution of pH 7.2. To study the interaction modes of BSA with the benzazoles, a stock solution of BSA (2.0 mg·mL^−1^) with a concentration of 31 µM was initially prepared in PBS at pH 7.2. To analyze the fluorescence quenching of BSA, solutions were prepared with a final volume of 3 mL with a fixed BSA concentration of 11 µM in PBS, pH 7.2. Benzazoles were added to these solutions at different concentrations (0–20 µM) from a stock solution in dimethylformamide. UV-Vis absorption and fluorescence emission spectra were obtained at 25 °C. The emission curves were obtained under excitation at 277 nm, relative to the absorption wavelength of BSA, using Exc/Em slits of 5.0/5.0 nm, respectively. The BSA association study was performed keeping the dye concentration constant (2 μM in PBS). In this study, different amounts of previously prepared BSA solutions (0–10 μM in PBS, pH 7.2) were added. The final solution was kept to rest for 1 h. The fluorescence spectra were obtained at 25 °C and under the excitation of each absorption maxima.

### 3.3. Molecular Docking

The 3D structure of BSA was obtained from the Protein Data Bank (http://www.rcsb.org/pdb, ID: 4f5s, accessed 1 October 2019) [66]. The Chimera 1.8 software [67] was used to remove chain B, water, and other molecules, and add hydrogen atoms to the BSA protein. The ligands were built in the software Avogadro 1.1.1 [68], following semi-empirical PM6 [69] geometry optimization using the program MOPAC2012 [70]. The ligands and proteins in the *pdbqt* format were generated by AutoDockTools, where the ligands were considered flexible (with PM6 charges) and the enzyme rigid (with Gasteiger charges) [71]. The AutoDock Vina 1.1.1 program was used for blind docking [72], with a gridbox of 92 × 62 × 86 and the coordinates x = 9.457, y = 23.359, and z = 98.149 (exhaustiveness of 150). Semi-flexible docking was performed for the region surrounding the Trp134 residue, with a gridbox of 30 × 30 × 35 and the coordinates x = 20.324, y = 33.690, and z = 97.801 (exhaustiveness of 150). The lateral chain of the residues Arg185, Lys136, Lys131, Trp134, Tyr137, and Tyr160, was considered flexible during docking to improve the interactions between compounds and proteins. As a model for the binding pose, the compound conformer with the lowest binding energy (ΔG) was selected from the semi-flexible docking experiment. The docking results were analyzed using the Accelrys Discovery Studio 3.5 software [73].

## 4. Conclusions

In summary, we report here the photophysical characterization of ESIPT fluorescent lipophilic benzazoles and their binding affinity towards BSA by BSA fluorescence quenching and molecular docking. In PBS solution (pH 7.2), these compounds produce absorption maxima in the UV region related to ^1^π–π* electronic transitions. The derivatives with short alkyl chains exhibited absorption maxima at higher energies, indicating that the alkyl chain influences the electronic properties in the ground state. In addition, compounds with longer alkyl chains demonstrated broad absorption in the visible region, which could be related to the aggregation of these compounds. The benzazoles are fluorescent, with the main emission band located in the blue–green regions with a large Stokes shift. The nature of the emission bands seems to be different depending on the hydrophobic portion of the molecule, once again indicating that the alkyl chains influence their photophysics. The fluorescence quenching of BSA in the presence of the benzazoles demonstrated a significant decrease in the fluorescence intensity without a significant red-shift. The *k*_q_ values of approximately 10^12^ M^−1^·s^−1^ indicate that the interactions proceed according to a static mechanism. In addition, higher association constant values (*K*_A_ ~10^4^ M^−1^) were obtained from the double logarithmic plot relating the fluorescence intensities from BSA and the benzazoles, suggesting a strong interaction with BSA. Association with BSA could be observed only for benzazole 12. The docking simulations indicate that compounds 13 and 16 bind closely to Trp134 in domain I and that the additional compounds were bound between domains I and III, and did not directly interact with Trp134. Finally, the obtained binding parameters indicate that all the studied benzazoles could be efficiently transported and biodistributed by BSA in the bloodstream. 

## Data Availability

Not applicable.

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
