# Peer review of "Interaction Study between ESIPT Fluorescent Lipophile-Based Benzazoles and BSA"

_molecules, 2021, doi:10.3390/molecules26216728_

Round 1

Reviewer 1 Report

The present paper describes electronic spectral features of the series of benzazoles and their binding behavior with BSA. The latter was studies by fluorescence quenching experiments and modeling simulation methods. The author revealed that the binding sites where the benzazoles interacted was affected by the alkyl-chain lengths, indeed the compounds 13 and 16 more effectively interacted with BSA by binding the domain I. I consider the present study would be publishable in Molecules because as such a study may contribute to application of the benzazole chromophores/fluorophores to photofunctional materials that show sensing a natural compound/protein. Before publication, I would request the authors to consider a few points.

(1) It would be good to add a comment on the reason why the authors used the benzazole derivatives. Although the benzazoles possess ESIPT characteristics, no relationship between their binding behavior and ESIPT is demonstrated.

(2) It would be highly desired that the authors investigate fluorescence spectra of the benzoazoles in the presence of BSA. Because the absorption band of benzazole-BSA conjugates was observed at ~350 nm (Figure 3), by excitation at the absorption band, one would expect to detect the ESIPT emission band in the BSA binding site. Such experiments would provide information of the binding sites by occurrence/manipulation of the ESPT. The current results only demonstrate fluorescence quenching by bound molecules that has been recognized.

(3) Line 24-27: ‘The docking simulations indicated that benzazoles with short alkyl chains bind closely to Trp134 in domain I of BSA, with similar binding poses and interactions. Moreover, benzazoles with longer lipophilic chains bind between domains I and III, and do not exhibit direct interactions with Trp134.’ The sentences are contradictory to ‘The docking simulations indicate that compounds 13 and 16 bind closely to Trp134 in domain I (line 260-261).’

(4) I would suggest the authors to express Stokes shift not by wavelength but by wavenumber: wavenumber relates with energy but wavelength does not show energy.  

Author Response

Reviewer #1

The present paper describes electronic spectral features of the series of benzazoles and their binding behavior with BSA. The latter was studied by fluorescence quenching experiments and modeling simulation methods. The author revealed that the binding sites where the benzazoles interacted were affected by the alkyl-chain lengths, indeed the compounds 13 and 16 more effectively interacted with BSA by binding the domain I. I consider the present study would be publishable in Molecules because as such a study may contribute to the application of the benzazole chromophores/fluorophores to photofunctional materials that show sensing a natural compound/protein. Before publication, I would request the authors to consider a few points.

(1) It would be good to add a comment on the reason why the authors used benzazole derivatives. Although the benzazoles possess ESIPT characteristics, no relationship between their binding behavior and ESIPT is demonstrated.

Answer: Thanks for the comment. Firstly, it was decided to use ESIPT based compounds due to their high photostability and absent inner-filter effect. Secondly, the different alkyl chains were also proposed in these compounds due to previous results from our research group using cyanines (J.  Org. Chem., 2014, 79, 5511-5522.), where we could observe that alkyl chain length was fundamental for the BSA detection in PBS. In addition, the use of ESIPT dyes and their well-known very large Stokes shift would be useful for association experiments, where we investigate the dye emission in presence of different amounts of the biomacromolecule. This feature could be useful since the dye emission will be redshifted about the BSA emission, minimizing some potential interference of the BSA emission into the dye one. However, during the manuscript preparation, it was decided to present only suppression studies. However, based on the manuscript revision and comments it was decided to present the association study with BSA. All this new data was measured before the lockdown (COVID-19), and that is why we are able to present it in this revision, since we still are facing severe restrictions to access our laboratory. To clarify this topic, and also to justify the use of these ESIPT based compounds, new association experiments were added and discussed and the justification of using such compounds was also presented in the revised version of the manuscript.

(2) It would be highly desired that the authors investigate fluorescence spectra of the benzazoles in the presence of BSA. Because the absorption band of benzazole-BSA conjugates was observed at ~350 nm (Figure 3), by excitation at the absorption band, one would expect to detect the ESIPT emission band in the BSA binding site. Such experiments would provide information on the binding sites by occurrence/manipulation of the ESPT. The current results only demonstrate fluorescence quenching by bound molecules that have been recognized.

Answer: Thanks for the comment and for the suggestion. Association experiments are now presented and discussed. In this way, an aditional author was added to the manuscript, which was the graduation student responsible for the association studies. All spectra can be found in the manuscript and as supplementary material. The respective discussions were also presented in this new version. Finally, BSA-binding parameters, such as quenching percentage, Stern–Volmer quenching constant, bimolecular quenching rate constant, binding constant, number of binding sites, and experimental Gibbs free energy were obtained and discussed from supression experiments.

(3) Line 24-27: ‘The docking simulations indicated that benzazoles with short alkyl chains bind closely to Trp134 in domain I of BSA, with similar binding poses and interactions. Moreover, benzazoles with longer lipophilic chains bind between domains I and III, and do not exhibit direct interactions with Trp134.’ The sentences are contradictory to ‘The docking simulations indicate that compounds 13 and 16 bind closely to Trp134 in domain I (line 260-261).’

Answer: Thanks for the comment. We totally agree with this observation. During the description of the docking results for the abstract, we got confused with the numbering of the compounds, leading us to a different conclusion from the one presented in the docking results. Please note that all data was revised and the sentence was rewrite to avoid misunderstandings.

(4) I would suggest the authors express Stokes shift not by wavelength but by wavenumber: wavenumber relates with energy but wavelength does not show energy.  

Answer: Thanks for the suggestion. All Stokes shifts are now presented as wavenumber in the revised version of the manuscript. The respective discussions were also revised concerning the Stokes shift.

Reviewer 2 Report

In the manuscript Kroetz et al reports a general method to determine binding of benzazole with BSA. The authors observe a large stoke shift in ultraviolent region that they link to proton transfer in excited state. They further supported their results with computational analysis. The manuscript is informative and is of interest to general readers of Molecules. The reviewers suggest some minor corrections to improve the article.

  1. Is the quenching visible in other aqueous buffers such as acetate buffer, phosphate buffer etc.
  2. What is the effect of different pH on the quenching?
  3. Are the results consistent with other albumin such as HSA, MSA and RSA?
  4. Figure 1. The exact concentration of the fluorophores should be mentioned.
  5. Figure 1 and 2 are missing y-axis scale.
  6. Line 310. It should be 1.0

Author Response

Reviewer #2

In the manuscript Kroetz et al reports a general method to determine binding of benzazole with BSA. The authors observe a large stoke shift in ultraviolent region that they link to proton transfer in excited state. They further supported their results with computational analysis. The manuscript is informative and is of interest to general readers of Molecules. The reviewers suggest some minor corrections to improve the article.

  1. Is the quenching visible in other aqueous buffers such as acetate buffer, phosphate buffer etc.

Answer: Thanks for the comment. That is a good question. Usually, we do not investigate these compounds in different buffered solutions than PBS (pH 7.2) since it is well-known that at these conditions, the studied compounds are present in their neutral species. However, depending on the buffer, the benzazoles can ionize and probably change the dye-BSA binding, which may affect the quenching but in our opinion, do not suppress it, as the interaction is more dependent on the hydrophobic part of the molecule.

  1. What is the effect of different pH on the quenching?

Answer: Thanks for the comments. Since the quenching is directly related to the dye-BSA binding, changes in the electronic structure of the dye could also affect this interaction. At the studied PBS (pH 7.2), we could not observe any ionization of the studied compounds (absence of red-shifted absorption bands higher than 380 nm – J. Phys. Chem. A 2020, 124, 288-299 and Photochem. Photobiol. Sci. 2017, 16, 840-844). However, we believe that at higher pH the respective zwitterionic species are present and will affect the dye-BSA binding, which for these compounds are generally related to π-π stacking, hydrophobic interactions, hydrogen bonds (H-bonds), π-H bond, and a cation-π interaction, as it could be observed by molecular docking.

  1. Are the results consistent with other albumin such as HSA, MSA and RSA?

Answer: Thanks for the comment. That is quite an interesting suggestion, and usually, our research group also investigates Human Serum Albumin for a better understanding of the interaction of the dye. However and unfortunately, we are still facing severe restrictions to back to the laboratory due to the COVID-19. At this point, we are not able to perform additional experiments (HAS, MSA, or RSA), than those already performed in this investigation. We believe that similar behavior could be obtained in HSA due to their similarity to BSA (76% identity and 88% similarity in its sequence with HSA - Biochim. Biophys. Acta 1594 (2002) 84-99). However, concerning Mouse and rabbit serum albumins (MSA and RSA, respectively), which present lower identity and similarity with BSA (Scientific Reports 2018, 8, 14648; Molecular Immunology 2012, 52, 174-182 and Comparative Biochemistry and Physiology Part C: Pharmacology, Toxicology and Endocrinology 1995, 112, 257-266), additional experiments should be performed to answer this interesting question.

  1. Figure 1. The exact concentration of the fluorophores should be mentioned.

Answer: Thanks for the comment. The exact concentration of the fluorophores is now presented in the revised version of the manuscript.

  1. Figure 1 and 2 are missing y-axis scale.

Answer: Thanks for the comment. The Figures are normalized, which means that the data is presented between 0 and 1. To clarify this topic, this range is now presented in the respective figures.

  1. Line 310. It should be 1.0

Answer: Thanks for the comment. This issue was revised.

Reviewer 3 Report

The authors synthesized 2-(2′-hydroxyphenyl) benzazole-based compounds with various alkyl chains and examined their absorption and fluorescence characteristics. Then, they studied the interaction with BSA through fluorescence quenching assays. Based on the Stern–Volmer relationship model, the parameters of interactions were calculated. Finally, the interaction was studied by molecular docking simulations. The docking simulations indicated that benzazoles with short alkyl chains bind closely to Trp134 in domain I of BSA, with similar binding poses and interactions. Moreover, benzazoles with longer lipophilic chains bind between domains I and III, and do not exhibit direct interactions with Trp134.

I feel that the studies are too preliminary and lack discussion on the results.

Although the authors studied interaction between benzazoles and BSA by two methods, the comparison was not performed. If the interaction mode differs by the length of lipophilic chains, fluorescence quenching assays should show some difference. It is difficult to understand the meaning of the description at the end of Results and Discussion, "These results agree with experimental results" The predicted thermodynamic data (ΔGbind) obtained by molecular docking demonstrated that the compounds with short carbon chains (12 and 15) exhibited the lowest ΔGbind. But, the tendencies of experimental ΔG values is different.

Therefore, I think that this manuscript is too preliminary for publication.

Author Response

Reviewer #3

The authors synthesized 2-(2′-hydroxyphenyl) benzazole-based compounds with various alkyl chains and examined their absorption and fluorescence characteristics. Then, they studied the interaction with BSA through fluorescence quenching assays. Based on the Stern–Volmer relationship model, the parameters of interactions were calculated. Finally, the interaction was studied by molecular docking simulations. The docking simulations indicated that benzazoles with short alkyl chains bind closely to Trp134 in domain I of BSA, with similar binding poses and interactions. Moreover, benzazoles with longer lipophilic chains bind between domains I and III, and do not exhibit direct interactions with Trp134.

I feel that the studies are too preliminary and lack discussion on the results.

Although the authors studied the interaction between benzazoles and BSA by two methods, the comparison was not performed. If the interaction mode differs by the length of lipophilic chains, fluorescence quenching assays should show some difference. It is difficult to understand the meaning of the description at the end of Results and Discussion, "These results agree with experimental results" The predicted thermodynamic data (ΔGbind) obtained by molecular docking demonstrated that the compounds with short carbon chains (12 and 15) exhibited the lowest ΔGbind. But, the tendencies of experimental ΔG values is different. Therefore, I think that this manuscript is too preliminary for publication.

Answer: Thanks for the evaluation of our study. We could observe that the interaction mode was affected by the length of lipophilic chains, leading to fluorescence quenching assays with some difference (quenching efficiency from 41-53%). In this sense, we believe that the quenching experiments were affected by the chemical structure (length of lipophilic chains). However, the driving force of this interaction is not only the alkyl chains but also the fluorophoric unit, since molecular docking also indicated, despite the hydrophobic interactions, π-π stacking, hydrogen bonds (H-bonds), π-H bond, and cation-π interactions. That is why we are not able to see a clear tendency to take only the length of lipophilic chains into account. In addition, the description at the end of Results and Discussion is nonsense and it was revised as suggested. We agree with the reviewer that different values were found experimentally and theoretically for the ΔGbind, with no clear tendency between the compounds obtained by supression studies, but both studies indicated spontaneous interaction with BSA with values of the same order of magnitude. Finally, we hope that all changes, as well as new experiments added to this revision, lead to a positive evaluation of our study.

We hope we have provided proper explanations for the questions presented in this letter. If there are questions, which remain unclear, please let us know.

Round 2

Reviewer 1 Report

The authors appropriately addressed the comments on the original manuscript. I would recommend this manuscript to be published in Molecules. Maybe the following minor changes will be made on proof correction.

  1. Author list: Rodembusch, --> Rodembusch
  2. Line 268: the fluorescence increased up 3.6 as well as redshifted the emission maxima 25 nm --> the fluorescence intensity increased up to 3.6 fold and the emission maxima red shifted by 25 nm

Reviewer 3 Report

The authors revised the manuscript according to my comment. Therefore, I agree to publish this one for publication.